# NMR-Based Metabolomics Reveals Effects of Water Stress in the Primary and Specialized Metabolisms of *Bauhinia ungulata* L. (Fabaceae)

**DOI:** 10.3390/metabo13030381

**Published:** 2023-03-03

**Authors:** Ana Júlia Borim de Souza, Fernanda Maria Marins Ocampos, Rafael Catoia Pulgrossi, Anne Lígia Dokkedal, Luiz Alberto Colnago, Inês Cechin, Luiz Leonardo Saldanha

**Affiliations:** 1Faculty of Sciences, São Paulo State University (UNESP), Bauru 17033-360, SP, Brazil; 2Embrapa Instrumentation, São Carlos 13560-970, SP, Brazil; 3Department of Statistics, Federal University of São Carlos (UFSCar), São Carlos 13565-905, SP, Brazil

**Keywords:** *pata-de-vaca*, water regimes, flavonoids, D-pinitol, diabetes

## Abstract

*Bauhinia ungulata* is a plant used in Brazilian traditional medicine for the treatment of diabetes. Phytochemical studies revealed flavonoids and the saccharide pinitol related to hypoglycemic activity of the *Bauhinia* species. To determine the effects of water deficit on ecophysiological parameter and metabolite fingerprints of *B. ungulata*, specimens were treated with the following water regimens under greenhouse conditions: daily watering (control), watering every 7 days (group 7D), and watering every 15 days (group 15D). Metabolite profiling of the plants subjected to water deficit was determined by LC-HRMS/MS. An NMR-based metabolomics approach applied to analyze the extracts revealed increased levels of known osmoprotective and bioactive compounds, such as D-pinitol, in the water deficit groups. Physiological parameters were determined by gas exchange in planta analysis. The results demonstrated a significant decrease in gas exchange under severe drought stress, while biomass production was not significantly different between the control and group 7D under moderate stress. Altogether, the results revealed that primary and specialized/secondary metabolism is affected by long periods of severe water scarcity downregulating the biosynthesis of bioactive metabolites such as pinitol, and the flavonoids quercetin and kaempferol. These results may be useful for guiding agricultural production and standardizing medicinal herb materials of this medicinal plant.

## 1. Introduction

*Bauhinia ungulata* L. (Fabaceae), popularly known as “*pata-de-vaca*” (cow’s paw) or “*mororó*” [1], is a common species in the Brazilian Savanna (Cerrado) [2]. *Bauhinia ungulata* is distributed in the north, center-west, southeast, and northeast regions of Brazil, and its leaves have been used in the treatment of diabetes and diarrhea as a traditional medicine by South American populations [1,3]. Phytochemical studies of the leaves have revealed several derivatives of flavonoids [4,5,6], beside stilbenoids [5], polyol [4], terpenoids, as well as sterols [5].

Specialized metabolites play an important role in the ecophysiological adaptation of plants to their environment [7,8]. Their biosynthesis and accumulation are modulated by a combination of genetic and environmental factors [9]. Biotic and abiotic factors commonly affect the production of various classes of metabolites [10]. Many studies have demonstrated the influence of abiotic climatic factors on plant metabolism promoted by temperature variations, UV irradiation, and drought [11]. Therefore, an adequate evaluation of how climatic factors affect the biosynthesis and accumulation of specialized metabolites can provide a valuable understanding of medicinal plant performance under climatic stress conditions [10,12].

Plants are affected by many abiotic factors that impact on both the primary and secondary metabolism of plants [11]. However, most studies to date have focused on either the primary or secondary metabolism, without investigating the complex interactions between these two pathways [13]. Additionally, while some studies have examined the effects of abiotic stress on both metabolisms, they have mainly focused on crop species. For example, Fazani, Sanches and Silva (2013) [14] evaluated the effects of water deficit on only the primary metabolism of *Bauhinia forficata* L., a native medicinal plant. On the other hand, Glaubitz et al. (2017) [15] assessed the effect of high temperature on both primary and secondary metabolism in rice plants (*Oryza sativa* L. (Poaceae)) and observed the involvement of the two metabolisms in an integrated manner as a response. Therefore, further studies are needed to understand how primary and secondary metabolism are impacted by abiotic effects in native species, especially those of medicinal interest. Such studies are crucial because native plant species are essential resources for traditional medicine, and their medicinal properties are often associated with their primary and or secondary metabolism.

To address these issues, we determined the effects of abiotic factors on both metabolisms of native Brazilian species. *Bauhinia ungulata* was selected for use in this study, owing to its medicinal use and wide distribution in the Southeast Cerrado [2,16]. The Cerrado climate is characterized by two predominant and well-marked seasons: a rainy period with the highest temperatures, and the dry season, with mild temperatures [17]. Therefore, drought stress is one of the keys limiting factors for the growth and physiological status of *B. ungulata*. Thus, our goal was to determine the effects of drought stress on *B. ungulata* metabolism and photosynthetic parameters in an integrated manner. The results of this study can provide an estimation and enable a better understanding of the physiological and phytochemical status of medicinal plants in environments with scarce water availability.

## 2. Materials and Methods

### 2.1. Plant Material and Cultivation

Specimens of 3-year–old sown *B. ungulata* were donated by the Jardim Botânico Municipal de Bauru, São Paulo, Brazil (22°20′30″ S, 49°00′30″ W). Voucher specimens were deposited at the UNBA Herbarium, under code number 6021.The identification of the species was confirmed by Prof. Dr. Vinicius Castro Souza. Plants were acclimated under natural photoperiod from October to December 2018 in a Van der Hoeven greenhouse, in the Faculty of Sciences, UNESP Bauru, São Paulo, Brazil (22°21′28″ S, 49°01′37″ W). The relative humidity was 40–60%, and minimum and maximum average temperatures were 22 and 31 °C, respectively. Photosynthetic active radiation (PAR) inside the greenhouse, was about 70% of the conditions in the open sky, due to polycarbonate transmission characteristics. 

### 2.2. Drought Conditions

Fifty-four specimens of *B. ungulata* were aleatorily divided into three experimental groups (n = 3) with different periods of water availability, under greenhouse conditions. Irrigation was controlled by drip during a total period of 90 days. The experimental groups were divided as follows: daily watered (control), watered every 7 days (group 7D), and every 15 days (group 15D) [14].

### 2.3. Sampling Procedure

The leaves of all specimens were harvested in fortnightly intervals (15, 30, 45, 60, 75, and 90 days). Leaf samples (10 g) were obtained by a cut at the base of the petiole, frozen at −80 °C, and immediately freeze-dried. After this, dried leaf samples were individually powdered using a knife grinder.

### 2.4. Preparation of the Extracts

Aliquots of 50 mg of the freeze-dried and powdered leaves of all specimens were transferred to 2 mL microtubes, with 1.5 mL of deuterated methanol (AcroSeal, Thermo Fisher Scientific, Leicestershire, UK) and water (Sigma-Aldrich, St Louis, MO, USA) (CD_3_OD:D_2_O) (7:3 *v*/*v*) containing 0.05% *w*/*w* of tetramethylsilane (TMS) (Cambridge Isotope Laboratories, Andover, MA, USA). The extraction was performed via ultrasonication under room temperature for 20 min. After that, the obtained extract was centrifuged at 65,856× *g* for 20 min, and the supernatant was collected. The supernatant obtained was pretreated in C18 solid-phase extraction cartridges (SPE) (Teknokroma, Finiesterre, Spain) (500 mg/3 mL) to remove lipophilic substances. The cartridge was activated with deuterated and pre-equilibrated CD_3_OD:D_2_O (7:3 *v*/*v*). The supernatant was loaded and eluted into the cartridges, and collected in vials. This process afforded the hydroalcoholic leaves extract of *B. ungulata.* An aliquot of 600 μL was transferred to a 5 mm NMR tube and analyzed by NMR [18].

### 2.5. UHPLC-ESI-HRMS Analysis

The liquid chromatography analysis was performed on an UltiMate 3000 UHPLC (Thermo Fisher, Waltham, MA, USA) system, interfaced with a Q exactive hybrid quadrupole-orbitrap mass spectrometer (Thermo Fisher Scientific, Waltham, MA, USA), using a heated electrospray ionization (HESI-II) source. The separations were achieved using an Acquity UPLC BEH C18 Premier column (2.1 × 100 mm, i.d., 130 Å, 1.7 µm), maintained at 60 °C. The mobile phase consisted of 0.1% formic acid in water (solvent A) and acetonitrile (solvent B), at a flow rate of 0.75 mL/min; a gradient elution was performed as follows: 5% to 50% of B in A for 4 min; 50% to 95% of B in A for 3 min; 95% of B in A for 1 min. The total running time was 8 min. The injection volume was 2 µL. The optimized HESI-II parameters were set as follows: source voltage, 3.5 kV; sheath gas flow rate (N_2_), 48 units; auxiliary gas flow rate, 11 units; spare gas flow rate, 2.0 units; capillary temperature, 300 °C; S-Lens RF Level, 55 units. The mass analyzer was calibrated using a mixture of caffeine, methionine–arginine–phenylalanine–alanine-acetate (MRFA), sodium dodecyl sulfate, sodium taurocholate, and Ultramark 1621, in an acetonitrile–methanol–water solution containing 1% formic acid, by direct injection. The data-dependent MS/MS events were performed on the three most intense ions detected in full-scan MS (Top3 experiment). The MS/MS isolation window width was 2 Da, and the normalized collision energy (NCE) was set to 20, 35 and 50 units. In data-dependent MS/MS experiments, full scans were acquired at a resolution of 35,000 fwhm (at *m*/*z* 200) and MS/MS scans at 17,500 fwhm, both with a maximum injection time of 50 ms. After being acquired in a MS/MS scan, parent ions were placed in a dynamic exclusion list for 3.0 s. The analysis was performed in both negative and positive ion modes.

### 2.6. UHPLC-HRMS2 Data Processing

The UHPLC-HRMS^2^ raw data were converted to mzXML using the MS converter (Proteowizard, University of Southern California, EUA), and processed using MZmine 2.10 [19] for peak detection, peak filtering, chromatogram construction, chromatogram deconvolution, isotopic peak grouping, chromatogram alignment, and gap filling. This preliminary processing resulted in 1579 features which were further filtered to a final peak list containing 704 features, having an associated data-dependent MS^2^ spectrum. This resulting list with 704 ions was exported as input for the generation of the molecular network (MS^1^ and MS/MS data).

### 2.7. Molecular Network Analysis and Computational Annotation

The molecular network (MN) was generated using the online Feature-Based Molecular Networking [20] workflow of the Global Natural Products Social Molecular Networking (GNPS) [21]. The MS^2^ spectra were then clustered with MS-cluster, with a parent mass tolerance of 0.02 Da and a fragment ion mass tolerance of 0.02 Da, to create consentaneous spectra, and consentaneous spectra containing less than two spectra discarded. A network was then created where edges were filtered to have a cosine score above 0.7 and more than 6 matched peaks. Further edges between two nodes were kept in the network if, and only if, each of the nodes appeared in each other’s respective top 10 most similar nodes. All matches kept between network spectra and library spectra were required to have a score above 0.7 and at least 6 matched peaks. The compounds in the MN were annotated by means of spectral comparison against the ISDB-UNPD (in silico database, created from the Universal Natural Products Database) spectral library, as described by Allard et al. (2016) [22]. The parameters used to compare the spectral dataset were as follows: parent mass tolerance, 0.05 DA; minimum cosine score, 0.1; returning top 5 candidates. Scripts for the ISDB-UNPD metabolite annotation method are available at the following address: https://github.com/oolonek/ISDB (accessed on 10 November 2022). Chemical classification of compounds was performed using the ClassyFire program, freely available at http://classyfire.wishartlab.com (accessed on 10 November 2022). 

### 2.8. ^1^H-NMR and 2D-NMR Experiments

^1^H NMR, ^1^H-^13^C HSQC, and ^1^H-^13^C HMBC experiments were performed at 300 K on a Bruker AVANCE III HD 600 NMR spectrometer (Bruker, Karlsruhe, Germany), operating at 14.1 T, observing ^1^H and ^13^C at 600.13 and 150.90 MHz, respectively, and equipped with a 5 mm multinuclear direct detection probe with z-gradient. A QC sample, composed of a pool of all samples, was used to calibrate the 90° pulse length, to determine the offset of the water signal for the water suppression, and to periodically monitor acquisition. Each ^1^H-NMR spectrum consisted of 256 scans, with the following parameters: 0.73 Hz/point; acquisition time (AQ), 1.36 s; relaxation delay (RD), 2.0 s; 90° pulse width (PW), 11.5 µs. Spectra were acquired with the ICON software (Durham, United Kingdom) for automation of acquisition parameters, using 1D NOESY-presat (*noesygppr1d*) pulse sequence on a spectral width of ~20 ppm. One-bond (HSQC) and long-range (HMBC) ^1^H-^13^C-NMR correlation experiments were optimized for average coupling constants ^1^*J*_(H,C)_ and ^LR^*J*_(H,C)_ of 140 and 8 Hz, respectively. Spectra were processed using the TopSpin3.5 software (Bruker). All ^1^H and ^13^C-NMR chemical shifts were observed in ppm related to TMS signal at 0.00 ppm as an internal reference, and an exponential line broadening of 0.3 Hz was applied. After Fourier transformation, spectra were manually phased, and baselines were automatically corrected. Metabolites were assigned based on the chemical shifts, signal multiplicities, and integrals, always in comparison to the literature [18].

### 2.9. Quantitative Analysis by ^1^H-NMR (qHNMR)

Non-overlapped integrals (*I*) of the signals at 2.22 ppm (proline), 3.83 ppm (D-pinitol), and 6.19 (quercetin and kaempferol), were used to calculate the relative concentration (RC) of the target metabolites in relation to the TMS (0.00 ppm) in all samplings and for all the biological replicates (n = 3), considering the number of hydrogens (*N*), giving rise to each signal in the molecule [23,24,25], as follows:*RC* = *I*x/*I*y .*N*y/*N*x

Results are expressed as ln of the relative concentration of each target metabolite.

### 2.10. Physiological Parameters Analyses

At fortnightly intervals (15, 30, 45, 60, 75, and 90 days), photosynthesis (*A*), stomatal conductance to water vapor (*g_s_*), transpiration (*E*), instantaneous water use efficiency (*A/E*), and intercellular CO_2_ concentration (*C_i_*), were measured in planta for all plants. The analyses were performed in the youngest, fully expanded leaf, using a portable infrared gas analyzer (LCpro, ADC, Hoddesdon, UK) [26]. Evaluations were performed between 8 and 10 a.m., inside the greenhouse, under ambient temperature, partial carbon dioxide pressure, and air–water vapor pressure. A photosynthetic active radiation (PAR) of 1000 μmol m^−2^ s^−1^ was supplied by a light unit, mounted on the top of the leaf chamber. The leaf was kept under this PAR unit until a steady-state rate of photosynthesis was achieved. Biomass production was evaluated by measuring dry leaf mass after the leaves being oven-dried at 60 °C. The mass obtained was expressed as g plant^−1^.

### 2.11. Statistical Analysis

Multivariate data analysis of the ^1^H-NMR spectra (53 samples) was conducted using the SIMCA software (version 14.1, Umetrics, Umeå, Sweden). For this purpose, 249 simple rectangular buckets of 0.04 ppm were calculated using the AMIX software with a special integration mode, and scaling to total intensity. The data set included the regions between 0.05 and 10 ppm, excluding methanol and residual water signal regions. Principal component analysis (PCA, 53 spectra × 250 variables) and orthogonal partial least squares discriminant analysis (OPLS-DA) were performed using Pareto scaling. The OPLS-DA model had 253 variables (250 X variables and 3 Y variables), and two predictive components were calculated.

Data of physiological parameters, as well as q-NMR data, were subjected to analysis of variance (ANOVA), and any contrast between the means was evaluated by a Tukey’s test at 5% level of probability, using the statistical package R (Foundation for Statistical Computing, Vienna, Austria). Normality tests were previously performed. The log in the E-base was made in the quantitative analysis data before all the statistical processes carried out in the eco-physiological parametric and q-NMR, to guarantee the normality of the residuals and ensure the validity of the model assumptions.

## 3. Results and Discussion

To obtain structural information on the metabolites present in the hydroalcoholic leaf extract of *B. ungulata*, a representative sample (pool of all extracts used as quality control samples for the metabolomics experimentation) was prepared and subjected to metabolite profiling via LC-HRMS and NMR. The results of these two approaches are presented below.

### 3.1. Metabolite Profiling via LC-HRMS

The MS/MS spectra of the quality control sample (QC) were recorded using UHPLC-orbitrap, which allowed us to perform a molecular network analysis. The preliminary MS data treatment yielded 441 and 823 features in the negative and positive ion modes, respectively. The positive ion mode results were used as the basis for further identification of the metabolites. The results were organized using the Global Natural Products Social Network platform (GNPS) to generate a unique molecular network (MN) (Figure 1a) based on the feature-based molecular networking workflow (FBMN) [21].

The annotation of compounds in the MN was performed via spectral comparisons with an in silico spectral database, following an approach previously described [21]. Briefly, the acquired MS^2^ spectra of each node from the entire MN were searched automatically against an in silico theoretical spectra database built from the Universal Natural Products Database (ISDB-UNPD).

The obtained results are visualized in an MN (Figure 1), where the clusters are colored based on the chemical class classification, and the size of the nodes is proportional to the MS peak area.

To identify structurally related metabolites with high relative abundance in the hydroalcoholic leaf extract of *B. ungulata* (Figure 1b), clusters were selected based on node size. The five largest nodes were clustered together and revealed the presence of 19 flavonoids (1–18, 31) (Table 1). These compounds were identified as aglycones or glycosylated flavanols, flavones, and isoflavones. 

Complementary to the LC-HRMS analysis, the hydroalcoholic extract was analyzed via NMR to enhance the cover of the metabolome and a fingerprint of the most abundant compounds in the extract.

### 3.2. Metabolite Profiling via NMR 

The hydroalcoholic leaf extract of *B. ungulata* (QC sample) was analyzed by NMR and 11 compounds (19–31) were directly identified. Chemical structures were identified by comparing NMR data with the literature (Figure 2 and Table 2). The identity of each compound was confirmed by HSQC 2D correlation maps through the ^1^H-^13^C correlations (see Appendix A). The identified compounds were herein classified as primary and specialized metabolites. 

Free amino acids, L-alanine (19), L-valine (20), L-aspartate (21), and L-proline (22) were identified in the region between 0.9 and 4.0 ppm of the ^1^H NMR spectrum (Appendix A and Figure 3, Appendix A). These compounds are primary metabolites, vital for the plant physiological processes, as they are the building blocks for bigger structures, such as proteins. 

Another class of primary metabolites, also identified herein (Appendix A), were the free carbohydrates α-D-glucose (23), ß-D-glucose (24), and sucrose (26), in addition to the polyol D-pinitol (25). These free carbohydrates are ubiquitously distributed in plants because they are essential features for plant cells, as they act as respiratory substrates for energy generation and biosynthesis of intermediary metabolites in specialized metabolism [1]. In the ^1^H NMR spectrum of mixtures, sugar signals are frequently superimposed on the region of 3 to 4 ppm, thus the information given by the hydrogen attached to the anomeric carbon is crucial to the structure identification, together with the 2D experiments, such as the HSQC (Appendix A) and HMBC ^1^H-^13^C correlation maps. The HMBC correlation map of 25 and 26 are shown in Appendix A, respectively, which highlight the long-distance ^1^H-^13^C correlations of the two polyols. Glucose and sucrose are the common sugars present in plants, while D-pinitol is a known inositol ether, also widely distributed in plants, especially Fabaceae [35]. D-pinitol was previously identified in *B. ungulata* [4], *B. holophylla* [36], and *B. variegata* leaves [37,38].

The specialized metabolites identified herein (Appendix A) were 4,4′-dihydroxybibenzyl (27), the gallic acid derivatives (28 and 29), kaempferol (30), and quercetin (31). HSQC and HMBC ^1^H-^13^C correlation maps are shown in Appendix A. The two most intense doublets (6.60 and 7.00 ppm) on the aromatic region of the ^1^H NMR spectrum were attributed to the stilbenoid (27). The intensity of these signals is due to the symmetry in both aromatic rings and the molecule, meaning that there are only two chemical environments for all eight aromatic hydrogens. The aliphatic portion of the (27) present integrals was compatible with the two-methylene present in this symmetric molecule derived from the phenylpropanoid pathway [39]. The gallic acid derivatives 28 and 29 were partially characterized and compared with the available literature data. The chemical shift of the carbonyl group was lower than the expected value [40] for the carboxylic acid of gallic acid, suggesting an ester substitution in the acidic portion. However, the connections with the substituents did not appear in the correlation maps, and the identification was inconclusive by NMR. For full characterization of compounds 28 and 29, further experiments after chromatographic separation would be necessary. The flavanols 30 and 31 present a similar substitution pattern on the rings A and C, and therefore, similar chemical shifts for the hydrogens attached to the carbons 6 and 8. Therefore, the substitution pattern on the ring B evidenced the two different aglycone structures, kaempferol and quercetin. 

Among all the substances identified by the NMR analyses, compounds 25 [4], 30 [4,5,6], and 31 [4,5,6] were already described in *B. ungulata*. Stilbenoids have previously been reported in this species [5]; however, compound 27 was identified for the first time in the leaves of *B. ungulata*. Of note, to our knowledge, seven of these primary substances (19, 20, 21, 22, 23, 24, and 26) had not been described in *B. ungulata*.

An ^1^H NMR spectrum of the QC sample was used as a fingerprint of the hydroalcoholic leaf extract of *B. ungulata*, to show the hydrogen signals attributed to each compound. Moreover, for a general overview, representative ^1^H NMR spectra of hydroalcoholic leaf extract of *B. ungulata* from the control group and two treatments—7 days without irrigation (7D) and 15 days without irrigation (15D)—are presented in Figure 2.

### 3.3. NMR-Based Metabolomics

The drought conditions within a greenhouse, and the sampling plan implemented in this study, allowed us to obtain leaf samples of 53 specimens of *B. ungulata.* One replicate of the 15D treatment did not survive the stress conditions after 90 days of the experiment. Therefore, chemical analysis was performed on 53 samples.

To assess the metabolite fingerprinting of the samples obtained from the specimens subjected to different irrigation periods, an optimized NMR-based metabolomics method was developed. To calibrate the 90° pulse length to offset the water signal and evaluate the efficiency of the metabolomics method and the NMR platform, a quality control sample was prepared from a pool of experimental samples. The quality control (QC) sample was systematically and regularly injected between the experimental samples, and used to validate the quality of the profile obtained [18]. Thus, based on QC sample validation, the reproducibility of the fingerprint (Appendix A) was found to be within an acceptable range, demonstrating instrument stability over the experiment.

### 3.4. Multivariate Data Analyses

The NMR data were processed and subjected to standard multivariate data analysis to investigate differences among the experimental groups. Briefly, we first employed an unsupervised method to analyze the variation in the data of the related groups. Therefore, principal component analysis (PCA) was performed using Pareto scaling, with statistical values of Rx^2^ (0.85) and Q^2^ (0.3). The PCA results provided good discrimination of all samples, indicating the presence of three main groups, as well as chemical differences among the experimental groups (Figure 4a). 

The PC1 separated samples according to the experimental group, explaining 37.9% of the variation. PC2 captured variation among samples from the same experimental group, explaining 18.2% of the variation. Indeed, a clear separation was observed between the control (negative PC1) and 15D (positive PC1) groups. As expected, group 7D was distributed along PC1, between the two other groups.

To identify features that mostly contributed to the discrimination of the groups revealed in the PCA, we proceeded to a supervised model [37]. Briefly, an OPLS-DA model was created (Figure 4b), classifying samples according to their experimental group. The variance explained by the R2X was 63%. The model presented an R2Y of 0.53, indicating high-quality discrimination (>0.5) and a Q2 of 0.35. The model calculated two predictive components expressing information from both X and Y, and one orthogonal in X components, which express information found only in X. On the score scatter plot, observations that were far from each other differed more than observations that were relatively close to each other. Accordingly, the model could partially distinguish the groups, as the 7D group was in the middle and partially superimposed with one of the two other groups (control and 15D). Specimens receiving irrigation every 15 days were clustered on positive axis one, while plants that were irrigated daily were clustered on both negative axes 1 and 2. The overlapping of group 7D with the other groups was acceptable because it was an intermediate treatment. Only two samples were placed inside the tolerance ellipse based on Hotelling’s T2, displaying an acceptable quality of the dataset, even with the expected variance of the biological replicates.

### 3.5. Identification of Stress Biomarkers for Drought

The analysis of the loading column plot (Figure 4c,d) allowed us to identify the features that contributed to the differentiation of the groups under different water regimens, and identify biomarkers for drought stress. The loading column plot of PC1 demonstrated that L-proline (22) and D-pinitol (25) were present at higher levels in treatments with lower water availability (7D and 15D). The carbohydrate derivatives, α-D-glucose (23), and β-D-glucose (24), were present at higher levels in specimens from group 7D, while sucrose (26) was present at higher levels in plants of the control group.

One of the main responses of drought-tolerant plants to stress is the transport of solutes (e.g., amino acids and carbohydrates) to the vacuoles, to regulate the balance between the cells’ osmotic potential and turgor [41]. Among the organic compounds that accumulate in foliar cells under stress due to water deficit, glucose is highlighted, owing to its osmoprotective activity [41,42] and its role in inducing the activation of many stress-responsive genes, acting as a signaling molecule during abiotic stress [43].

A decrease in carbohydrates is observed in many plant species under drought stress [44]. This can be explained by the decrease in the photosynthetic rate and the altered distribution and metabolism of the carbon in plants due to water stress, which leads to energy depletion and a decreased yield. Sucrose is one of the main products of photosynthesis, and when drought limits this activity, there is a tendency for the levels of this carbohydrate to decrease [45]. In such conditions, the stored sucrose must be used, which may markedly reduce its levels [46].

The signals of chemical shifts attributed to phenolic compounds, 4,4′-dihydroxybibnzyl (27), gallic acid derivatives (28 and 29), kaempferol (30), and quercetin (31), were decreased in specimens from the 7D and 15D groups compared to the control group. The control group, which received daily irrigation, did not show any signs of stress based on the chemical fingerprint obtained, due to scarce irrigation, as expected.

The above finding suggests that treatment groups with lower water availability (7D and 15D) present different contents of carbohydrates, phenolic compounds, and amino acids in relation to the control, which can be used as drought stress biomarkers.

### 3.6. Quantitative Analysis by NMR

To further investigate the contents of the selected markers in the treatment groups, ^1^H-NMR quantitative analysis was performed, which allowed us to estimate their relative concentrations and better understand the chemical differences in their fingerprint profiles. Briefly, the signals referring to the identified markers of drought in the ^1^H spectra were selected (Figure 5). For the amino acids, the signal (2.22 ppm) of L-proline (22) was selected; for the polyols, a signal (3.83 ppm) of D-pinitol (25) was selected; and for the phenolic compounds, a coincident signal (6.19 ppm) of kaempferol (30) and quercetin (31) was selected.

The relative concentration of L-proline (22) (Figure 5a) showed a significant difference (*p* > 0.05) among all groups during the 90-day experiment. Both treatment groups (7D and 15D) presented higher values than the control group. The values for L-proline (22) increased with higher periods of water scarcity. For example, the 7D group presented two elevated units, while the 15D group had 3.8 units in relation to the control group that received daily irrigation. These results indicate that water stress increased with periods of water scarcity, according to the proline content [47,48].

Proline is a low-molecular-weight amino acid, that is putatively the first solute to concentrate in plant vacuoles. Proline has also been widely highlighted as a biomarker of stress due to water deficit [49]. Besides its osmoregulatory properties, the literature is very rich in describing other important functions in maintaining the integrity of biological membranes; stabilizing protein structures, carbon, and nitrogen storage; and balancing the redox status in plants in low-water-availability environments [50].

The relative concentration of D-pinitol (25) (Figure 5b) revealed a significant difference (*p* < 0.05) between the control group and the other two groups (7D and 15D). The 7D and 15D groups presented higher amounts of D-pinitol (25), with no significant difference between them. D-pinitol is a common cyclitol present in plants belonging to Fabaceae [35]. Some studies have reported increased concentrations of D-pinitol in response to osmotic stress conditions due to water deficit [46]. Cyclitols are known as osmoprotective solutes that can maintain turgor pressure at a lower hydric potential, conferring osmotic adjustment to plants [46]. Their diverse functions have been attributed to the presence of hydroxyl groups, which can replace water by establishing hydrogen bonds during limited water availability. Thus, these solutes can preserve enzymatic activities and the integrity of cell membrane structures [51,52]. This may explain the accumulation of D-pinitol (25) with *B. ungulata* specimens that were subjected to seven and 15 days of water scarcity.

Interestingly, D-pinitol has hypoglycemic properties [53] and is found in many medicinal plants used to treat diabetes [54]. Navarro et al. (2020) [55] demonstrated that male rats treated with D-pinitol showed pancreatic protection by decreasing insulin secretion. Diabetic rats treated with *Mesembryanthemum crystallinum* L. (Aizoaceae) extract, which is rich in D-pinitol, also had effective control of blood glucose levels [56]. D-pinitol was also found in higher concentrations, and was related to one of the active principles of *Bauhinia holophylla* (Fabaceae) leaf extracts, for diabetes treatment [36].

The relative concentrations of the flavonoids, kaempferol (30), and quercetin (31) did not show significant differences (*p* < 0.05) in any of the treatments over the 90-day experiment (Figure 4c). These results indicate that the content of phenolic compounds, as depicted by the concentration of the given compounds, was not altered by water scarcity for periods of 7 and 15 days over 90 days.

In contrast to the results obtained herein, several studies have demonstrated the accumulation of phenolic compounds in response to water stress [57]. Mechri et al. (2020) [58] reported the accumulation of phenolics in *Olea europaea* L. (Oleacea) subjected to water deficit. For *Chrysanthemum morifolium* Ramat. (Asteraceae), the authors observed an increase in flavonoids in response to water stress [59]. In *Oenanthe stolonifera*, Wall. (Apiaceae), an increase of 43% in phenolic content of the leaves was observed upon moderate water deficit [60].

Phenolics accumulate in plants to prevent water stress damage as a non-enzymatic antioxidant defense mechanism [61]. However, the biosynthesis of phenolic compounds is highly dependent on the amount of carbon fixed in plant leaves [62]. During water scarcity situations, photosynthetic activity tends to be reduced, leading to the lower production of photoassimilates needed for the biosynthesis of such complex compounds [63]. 

Stress factors tend to limit photosynthesis, and the photosynthetic capacity of a plant correlates negatively with the concentration of protective compounds [26]. For example, water loss through leaves is an unavoidable cost to acquire CO_2_ for photosynthesis. Plants that do not prioritize growth save water and simultaneously accumulate phenolic compounds for defense. However, these compounds can act as biochemical jokers, because they can also serve as a source of carbon and energy for growth [64].

We believe that these results can be explained in part by the prioritization of polyol accumulation, which is a substantial sink for carbon in the photosynthetic cycle [35]. Therefore, this could affect the availability of the carbon building blocks for phenolics. This competition for carbon can hinder the accumulation of phenolic compounds in the plant, as they are complex molecular structures and demand a substantial and diverse source of photo-assimilates for their synthesis [65,66].

The competition for carbon and the lower photosynthesis rate (see Figure 6) observed in our study, as discussed below, may explain the maintenance of the levels of these substances in the groups under water scarcity, similar to the control group. Neither the drought periods and/nor the duration of the experimentation implemented in this study were sufficient to activate mechanisms that would result in phenolic accumulation.

These results indicate that *B. ungulata* may have a primary response to drought and the accumulation of osmoregulatory compounds, such as amino acids and polyols. 

Notably, 45% of the world’s agricultural area is subjected to continuous or frequent drought conditions [67]. To date, few plants have been identified to produce osmoprotective solutes that allow them to resist such stress [68]. The ability to resist drought stress is an important feature for breeding programs of plants, with economic importance [45]. Moreover, these features are desirable for agronomic productivity, to fabricate plant-based products for pharmacological purposes.

### 3.7. Physiological Parameters

The degree of stomatal opening, given by *g_s_*, shows the gas exchange in the leaves, and controls the photosynthesis and transpiration of the plant. The values obtained for stomatal conductance (Figure 6a) shows that *g_s_* did not differ between 7D and 15D. However, the stomatal conductance was markedly reduced in both groups compared to the control group after 60 days, and remained reduced until the end of the 90-day experiment. The closing of the stomata to reduce water loss is considered one of the first responses of plants under water stress, as it prevents water loss, mainly through transpiration. The values for transpiration (Figure 6b) of plants in the 7D and 15D groups were statistically different relative to those of the control. As expected, transpiration (E) was significantly reduced after 15 days for the 15D group and after 30 days for the 7D group in relation to the control group, and remained reduced until the end of the 90-day experiment, indicating that the drought stress continued.

When water stress persists, photosynthesis may be limited by stomatal and/or non-stomatal effects (e.g., adenosine triphosphate and/or rubisco activity) [46]. As intracellular carbon concentration (*C_i_*) is affected by *g_s_* and is closely related to photosynthesis, we further analyzed *C_i_*. The values obtained for *C_i_* (Figure 6c) show that there was no significant difference in the intercellular CO_2_ concentration between treatments and control throughout the experimental period. The photosynthesis (*A*) analysis (Figure 6d) showed that *A* decreased at different levels, depending on the degree of water stress. Particularly, water scarcity significantly reduced photosynthesis in the 15D group, while the 7D group was especially affected after 60 days, which remained until the end of the experiment. Altogether, these results suggest that stomatal closure is not the main effect responsible for the reduction in photosynthesis, as *C_i_* remained at the same levels as in control plants, despite the decrease in *g_s_*. Therefore, in our study, 7D and 15D of water restriction may have caused a gradual shift to the non-stomatal limitation of photosynthesis, which reduced the demand for CO_2_.

Water use efficiency (WUE) represents the ultimate performance of water consumption and yield, determining the capacity of plants for water-saving and productivity. Some studies have demonstrated that different species improve WUE under water stress [69]. A more conservative use of water results in a higher WUE. This is considered a mechanism to improve resource utilization efficiency, and consequently, productivity under water scarcity. Our data showed that WUE was affected in a different manner among the experimental groups and at different time points (Figure 6e). For example, group 7D maintained similar values of instantaneous water use efficiency (A/E) as the control group until 60 days of experimentation. However, at 75 days, a significant difference was found compared to the control, with a decrease in A/E values. On the other hand, the 15D group was affected early in the experiment and showed a significant reduction in A/E values for almost the entire period of the experiment compared to the control group. Such finding suggests that WUE decreases significantly under severe water stress caused by 15D periods of water scarcity, but is not significantly affected under moderate stress, as observed for the 7D group. 

Importantly, the oscillations could be explained by air humidity variations during the experimentation period, as this environmental factor was not controlled during the measurements. Furthermore, the values of *A* were equal to those obtained for *E*, resulting in no difference in WUE.

The growth rate of plants under drought stress and physiological processes may be affected [70]. Our results show a significant decrease in the dry-mass yield (Figure 6f) of the plants in group 15D compared to plants in the 7D and control groups, especially between 45 and 75 days of the experiment. This result indicates that 15 days of water scarcity reduces *B. ungulata* growth, while moderate water stress does not affect biomass production. Water deficit can affect growth rates because cellular division and elongation are affected by the availability of carbon from photosynthesis [71]. This finding agrees with our results, where a lower biomass production was observed in specimens in the group that had a lower water supply.

This result indicates that specimens, with irrigation every 7 days, were not threatened regarding the biomass development of *B. ungulata.* In addition, the data obtained for the A/E (Figure 6e) could explain the dry-mass production observed in group 7D, as its results did not show a statistical difference relative to the control group.

## 4. Conclusions

The stress promoted by water deficit is generally considered an issue in agriculture because it causes serious income losses. In this context, medicinal plants that grow in dry climates generally produce higher concentrations of active substances than the same species growing in moderate climates. By identifying the threshold that allows for normal plant development, while also stimulating higher concentrations of pharmacologically active compounds, farmers can produce a lucrative product without compromising quality. The results of this study demonstrated that the interruption of irrigation of *B. ungulata* plants triggers the production of metabolites that may help the plant to survive stress conditions. Our data revealed that water supply every 15 days for long periods may cause damage to the plants. However, *B. ungulata* displayed tolerance to mild drought conditions, similar to the situation simulated by treatment 7D. Irrigation every 7 days did not significantly affect the ecophysiological parameters or production of specialized metabolites, which are usually responsible for pharmacological activities. In contrast, the mild stress caused by treatment with 7D enhanced the concentrations of primary metabolites, such as proline and D-pinitol, which exhibit osmoregulatory activity. Given the reported hypoglycemic activity of D-pinitol in the literature, this result is of special interest for the medicinal exploitation of this species. Hence, the interruption of irrigation of *B. ungulata* plants may be recommended not only to lower the production cost and save water, but also enhance the quality of the medicinal herb raw material. However, further pharmacological research on this subject is required.

## Figures and Tables

**Figure 1 metabolites-13-00381-f001:**
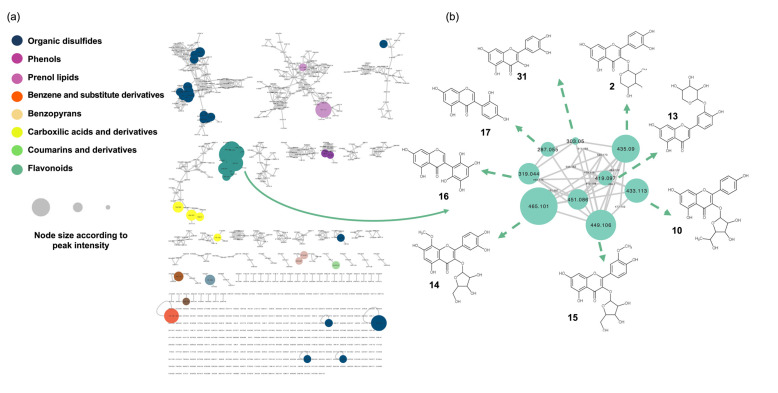
Molecular network analysis of the hydroalcoholic leaf extract of *Bauhinia ungulata.* (**a**) Color of the node indicates the chemical class of the compound. The node size is set according to the peak area of the mass signal. (**b**) Selected cluster, containing flavonoid derivatives. The chemical structure of the dereplicated compounds is given.

**Figure 2 metabolites-13-00381-f002:**
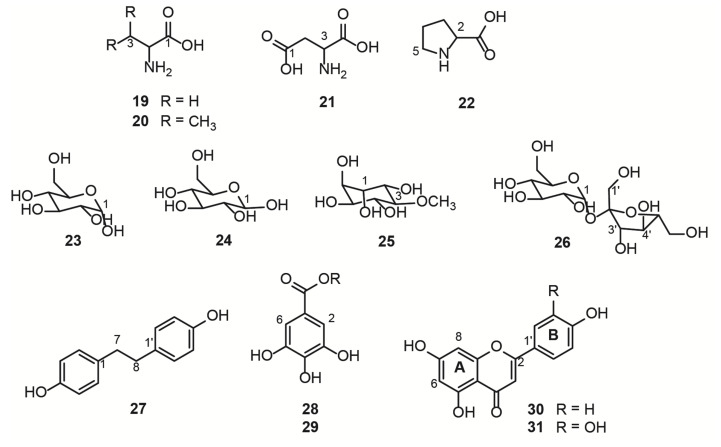
Structure of compounds identified by NMR in the hydroalcoholic leaf extract of *Bauhinia ungulata*.

**Figure 3 metabolites-13-00381-f003:**
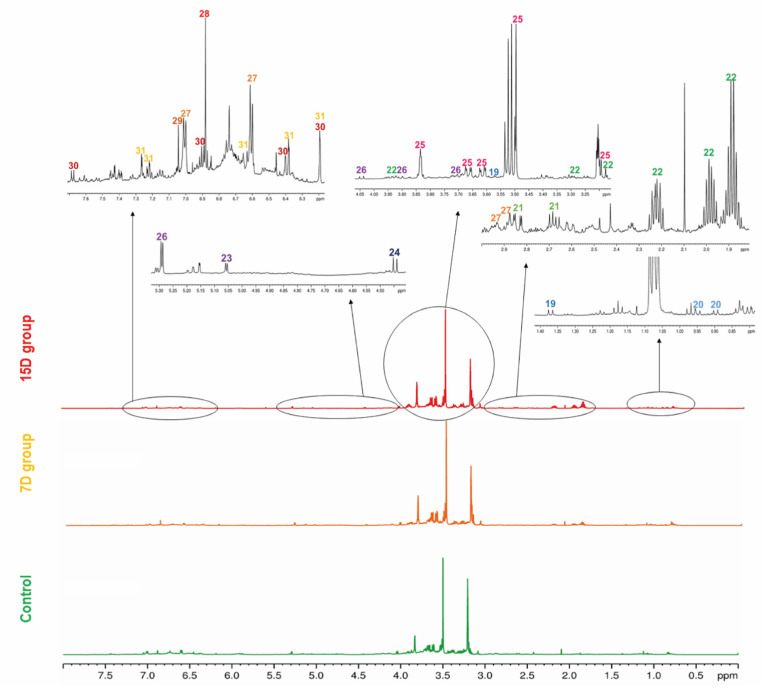
Representative ^1^H-NMR profile of the hydroalcoholic leaf extract of *Bauhinia ungulata* specimens with daily irrigation (control—green), one irrigation every 7 days (7D—orange), and one irrigation every fifteen days (15D—red). The expansion of the spectra regions showing the main signals of each compound (19–31) is presented. The color of the number indicates the class of the compounds according to Classyfire and mapped in the molecular network (see Figure 1).

**Figure 4 metabolites-13-00381-f004:**
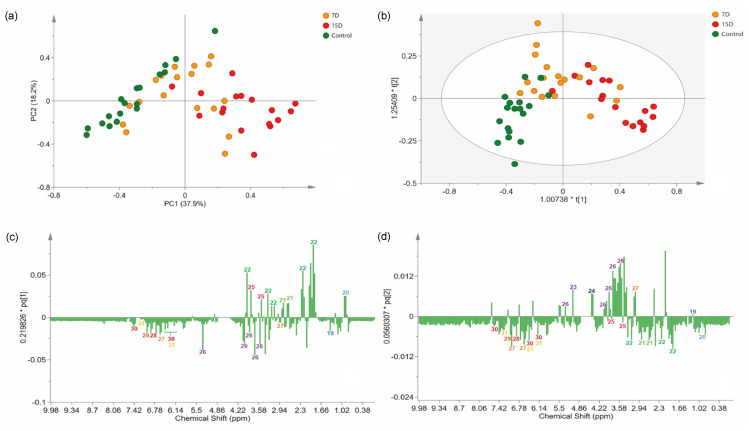
Multivariate analysis of ^1^H-NMR data of the hydroalcoholic leaf extract of *Bauhinia ungulata* specimens submitted to different water irrigation periods. (**a**) Score scatter plot of principal component analysis (PCA), showing three groups of samples according to water irrigation periods. (**b**) Orthogonal projections of latent structure discriminant analysis (OPLS-DA) score plot. (**c**,**d**) Loadings plot from the OPLS-DA, displayed as columns. Numbers in the columns indicate the identified features. The color of the number indicates the class of the compounds according to Classyfire and mapped in the molecular network (see Figure 1).

**Figure 5 metabolites-13-00381-f005:**
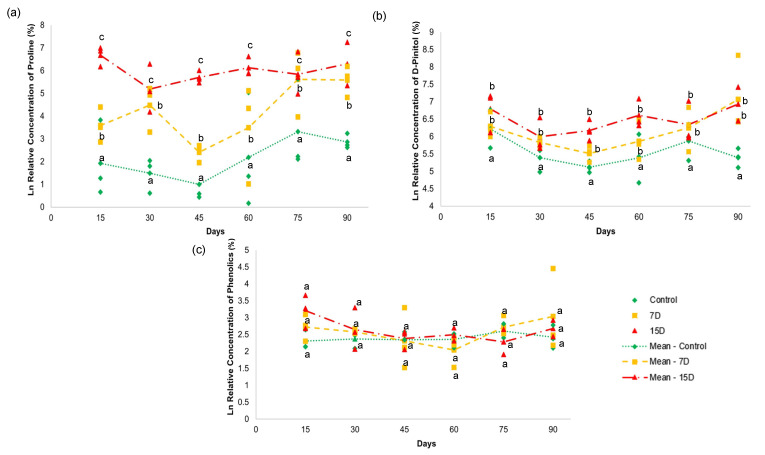
NMR quantitative analysis of (**a**) proline (22), (**b**) D-pinitol (25), (**c**) and total phenolics in the hydroalcoholic leaf extract of specimens of *Bauhinia ungulata* under daily irrigation (control—green), one irrigation every 7 days (7D—orange) and one irrigation every fifteen days (15D—red). The data are expressed as a log (ln) of the relative concentration of each compound. Different letters indicate significant difference (*p* < 0.05).

**Figure 6 metabolites-13-00381-f006:**
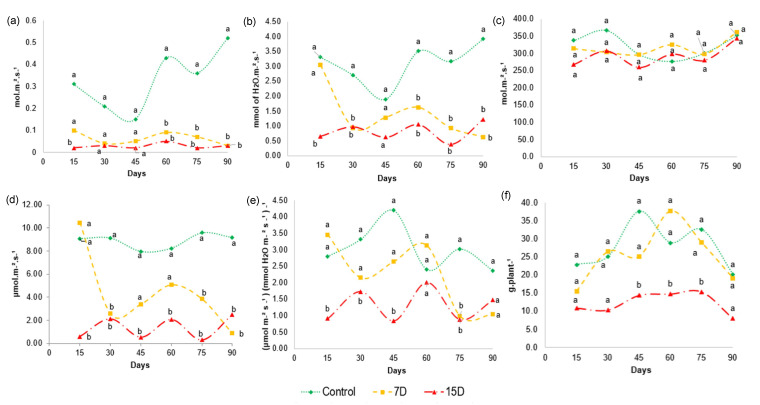
Gas exchange characteristics of *Bauhinia ungulata* during the 3-month experiment. (**a**) Stomatal conductance—*g_s_*, (**b**) transpiration—*E*, (**c**) intercellular CO_2_ concentration—*C_i_*, (**d**) photosynthesis—*A*, (**e**) instantaneous water use efficiency—*A/E* and (**f**) biomass production. (Tukey’s test; n = 3; *p* < 0.05). Values sharing the same letters are not significantly different at 5% significance level.

**Table 1 metabolites-13-00381-t001:** Identification of compounds in the hydroalcoholic leaf extract of *Bauhinia ungulata* by UHPLC-HRMS^2^ analysis in the positive mode. The InChIKey (International Chemical Identifier) for the annotated metabolites is presented.

ID	Annotation (Correspondent Compound in Database *)	Molecular Formula (Neutral)	*m*/*z* (M + H)^+^	Retention Time (min)	Chemical Class **	Partial InChIKey
1	Quercetin-3-*O*-*β*-*L*-Rhamnofuranoside	C_21_H_20_O_11_	449.1055	1.61	Flavonoids	OEKUVLQNKPXSOY
2	Quercetin-3-*O*-Arabinopyranoside	C_20_H_18_O_11_	435.0896	1.55	Flavonoids	PZZRDJXEMZMZFD
3	[(3R,4R,5R,6S)-6-[2-(5,7-Dihydroxy-4-oxochromen-2-yl)-4,5-dihydroxyphenoxy]-4,5-dihydroxyoxan-3-yl] acetate	C_22_H_20_O_12_	477.1023	1.84	Flavonoids	TUOJCDPSHAVURV
4	Quercetin 7-Rutinoside	C_27_H_30_O_16_	611.1589	1.38	Flavonoids	IVTMALDHFAHOGL
5	Tagetiin	C_21_H_20_O_13_	481.0965	1.23	Flavonoids	YUANNBKEZDNSIV
6	3′,4′,5,7-Tetrahydroxyflavone-4′-*O*-[α-L-Rhamnopyranosyl-(1→6)-*β*-*D*-glucopyranoside]	C_27_H_30_O_15_	595.1625	1.47	Flavonoids	WDQNUWOVEJHDOP
7	Mearnsitrin	C_22_H_22_O_12_	479.1158	1.65	Flavonoids	NAQNISJXKDSYJD
8	Tricetin 3-Glucoside	C_21_H_20_O_12_	465.1012	1.38	Flavonoids	XYILCYMQHZSECK
9	Tricetin	C_15_H_10_O_7_	303.0497	1.55	Flavonoids	ARSRJFRKVXALTF
10	Kaempferol 3-*O*-*α*-*L*-rhamnofuranoside	C_21_H_20_O_10_	433.1127	1.81	Flavonoids	FFFIPDPCGREKEW
11	Myricetin 3-*O*-*α*-Arabinofuranoside	C_20_H_18_O_12_	451.0856	1.35	Flavonoids	OXJKSVCEIOYZQL
12	Nitensoside B	C_28_H_32_O_16_	625.1730	1.55	Flavonoids	MTUPEWBIUKFRBD
13	Luteolin 3′-xyloside	C_20_H_18_O_10_	419.0966	1.74	Flavonoids	ZUMPYZVELBOZDM
14	Corniculatusin 3-*α*-*L*-arabinofuranoside	C_21_H_20_O_12_	465.1004	1.62	Flavonoids	DYNQYIRMFWJOJH
15	Isorhamnetin 3-O-*α*-*L*-arabinofuranoside	C_21_H_20_O_11_	449.1079	1.82	Flavonoids	OOZLPFOTSYKMTJ
16	2′,3′,5,5′,6′,7-Hexahydroxyisoflavone	C_15_H_10_O_8_	319.0437	1.38	Isoflavones	XFCGHCBRDUZOSI
17	2′-Hydroxygenistein	C_15_H_10_O_6_	287.0546	1.81	Isoflavones	GSSOWCUOWLMMRJ
18	4′,5,6,7,8-Pentahydroxyisoflavone-8-Me ether, 6-*O*-*α*-*L*-rhamnopyranoside	C_22_H_22_O_11_	463.1221	1.85	Isoflavones	VEDUBLIYMIMISG
31	Quercetin	C_15_H_10_O_7_	303.0498	2.09	Flavonoids	REFJWTPEDVJJIY

* Database: Universal Natural Products Database (UNPD). ** ClassyFire classification.

**Table 2 metabolites-13-00381-t002:** References reporting previous NMR data of the metabolites identified in the hydroalcoholic leaf extract of *Bauhinia ungulata*.

ID	Compound	Reference	Solvent
19	Alanine	[27]	D_2_O
20	Valine	[27]	D_2_O
21	Aspartate	[28]	D_2_O
22	Proline	[29]	D_2_O
23	*α*-D-glucose	[30,31]	CD_3_OD/D_2_O
24	*β*-D-glucose	[30,31]	CD_3_OD/D_2_O
25	D-pinitol	[32]	D_2_O
26	Sucrose	[30,31]	CD_3_OD/D_2_O
27	4,4′-Dihydroxybibenzyl	[33]	acetone-d_6_
28	Gallic acid derivative I	n.a. *	--
29	Gallic acid derivative II	n.a. *	--
30	Kaempferol	[34]	CD_3_OD
31	Quercetin	[34]	CD_3_OD

* n.a. = not avalaible.

## Data Availability

Not applicable.

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
