# Peer review of "NMR-Based Metabolomics Reveals Effects of Water Stress in the Primary and Specialized Metabolisms of *Bauhinia ungulata* L. (Fabaceae)"

_metabolites, 2023, doi:10.3390/metabo13030381_

Round 1

Reviewer 1 Report

The manuscript by de Souza et al. describes a nicely executed study. I have no major issues with it, only a ew minor points to be addressed as follow:

- line 91: What amount of leaf material (weight) was harvested for extraction?

- line 102: use `extract` instead of `solution`

- line 103: use `supernatant` instead of `solution`

- line 105: What was the solvent used in SPE to remove lipophilic compounds?

- lines 231-245: I feel this part should go into the Materials and methods section.

Author Response

Please, see the enclosed file. 

Reviewer 2 Report

Comments for the Authors

In the manuscript, authors have given an exhaustive account of the “NMR-based metabolomics reveals effects of water stress in the 2 primary and specialized metabolisms of Bauhinia ungulata L. 3 (Fabaceae)”. The authors have put in good amount of effort, and the work is appreciable. This paper deals with an important and interesting topic. The authors present their data in a novel approach and comprehensible manner making it an interesting subject for plant breeders. The experimental design and the methods used are appropriate, and the conclusions are well-supported by the data. Overall, I consider that the paper adopts new research methods and makes an excellent contribution to this research field; the results and discussion sections are well presented. I recommend article to accept for publication after minor revision. For the better readability of the manuscript I would like to recommend some changes and improvements.

1. First time mention the complete name and later use an abbreviation throughout the manuscript.

2. There are several mistakes in the whole manuscript, please add or remove extra characters, and check again.

3. The logic of research progresses should be adjusted and improved in the Introduction section.  

4. Please make sure that all the units must be in the same format.

5. English should be carefully checked and improved. The paper suffers from poor grammatical sentences and mistakes.
